# Comparative Study of the Effect of Radiation Delivered by Lutetium-177 or Actinium-225 on Anti-GD2 Chimeric Antigen Receptor T Cell Viability and Functions

**DOI:** 10.3390/cancers16010191

**Published:** 2023-12-30

**Authors:** Quaovi H. Sodji, Matthew H. Forsberg, Dan Cappabianca, Caroline P. Kerr, Lauren Sarko, Amanda Shea, David P. Adam, Jens C. Eickhoff, Irene M. Ong, Reinier Hernandez, Jamey Weichert, Bryan P. Bednarz, Krishanu Saha, Paul M. Sondel, Christian M. Capitini, Zachary S. Morris

**Affiliations:** 1Department of Human Oncology, University of Wisconsin School of Medicine and Public Health, Madison, WI 53792, USA; cpkerr@wisc.edu (C.P.K.); ashea6@wisc.edu (A.S.); pmsondel@humonc.wisc.edu (P.M.S.); zmorris@humonc.wisc.edu (Z.S.M.); 2Department of Pediatrics, University of Wisconsin School of Medicine and Public Health, Madison, WI 53792, USA; mhforsberg@wisc.edu (M.H.F.); ccapitini@pediatrics.wisc.edu (C.M.C.); 3Carbone Cancer Center, University of Wisconsin-Madison, Madison, WI 53792, USA; hernandez6@wisc.edu (R.H.); jweichert@uwhealth.org (J.W.); bbednarz2@wisc.edu (B.P.B.); ksaha@wisc.edu (K.S.); 4Department of Biomedical Engineering, University of Wisconsin-Madison, Madison, WI 53706, USA; dcappabianca@wisc.edu (D.C.); sarko@wisc.edu (L.S.); 5Department of Radiology, University of Wisconsin School of Medicine and Public Health, Madison, WI 53792, USA; 6Department of Medical Physics, University of Wisconsin School of Medicine and Public Health, Madison, WI 53792, USA; dadam3@jh.edu; 7Department of Biostatistics and Medical Informatics, University of Wisconsin School of Medicine and Public Health, Madison, WI 53792, USA; eickhoff@biostat.wisc.edu (J.C.E.); irene.ong@wisc.edu (I.M.O.); 8Department of Obstetrics and Gynecology, University of Wisconsin School of Medicine and Public Health, Madison, WI 53792, USA

**Keywords:** chimeric antigen receptor, targeted radionuclide therapy, Lutetium-177, Actinium-225, neuroblastoma, melanoma

## Abstract

**Simple Summary:**

Low-dose radiation delivered by radionuclides stimulates an immune response against cancer. We hypothesize that this type of low-dose radiation can potentiate chimeric antigen receptor (CAR) T cell therapy against solid tumors. Before evaluating the combination of these therapies in vivo, we aim to determine the impacts of this type of low-dose radiation on CAR T cell viability and functions to guide the selection of the type of radionuclide (actinium-225 or lutetium-177), the dose of radiation (1, 2 or 6 Gy) and how to best sequence their administration. It follows that increasing the radiation dose results in lower CAR T cell viability while enhancing their killing potential to the same extent. At a similar dose, radiation delivered by actinium-225 is more toxic to CAR T cells than lutetium-177. This suggests that 1 or 2 Gy delivered by lutetium-177 may be optimal for in vivo combination studies with CAR T cells.

**Abstract:**

Background and purpose. Chimeric antigen receptor (CAR) T cells have been relatively ineffective against solid tumors. Low-dose radiation which can be delivered to multiple sites of metastases by targeted radionuclide therapy (TRT) can elicit immunostimulatory effects. However, TRT has never been combined with CAR T cells against solid tumors in a clinical setting. This study investigated the effects of radiation delivered by Lutetium-177 (^177^Lu) and Actinium-225 (^225^Ac) on the viability and effector function of CAR T cells in vitro to evaluate the feasibility of such therapeutic combinations. After the irradiation of anti-GD2 CAR T cells with various doses of radiation delivered by ^177^Lu or ^225^Ac, their viability and cytotoxic activity against GD2-expressing human CHLA-20 neuroblastoma and melanoma M21 cells were determined by flow cytometry. The expression of the exhaustion marker PD-1, activation marker CD69 and the activating receptor NKG2D was measured on the irradiated anti-GD2 CAR T cells. Both ^177^Lu and ^225^Ac displayed a dose-dependent toxicity on anti-GD2 CAR T cells. However, radiation enhanced the cytotoxic activity of these CAR T cells against CHLA-20 and M21 irrespective of the dose tested and the type of radionuclide. No significant changes in the expression of PD-1, CD69 and NKG2D was noted on the CAR T cells following irradiation. Given a lower CAR T cell viability at equal doses and an enhancement of cytotoxic activity irrespective of the radionuclide type, ^177^Lu-based TRT may be preferred over ^225^Ac-based TRT when evaluating a potential synergism between these therapies in vivo against solid tumors.

## 1. Introduction

Chimeric antigen receptor (CAR) T cell therapy represents one of the three forms of adoptive T cell transfer therapy that has revolutionized cancer immunotherapy [1]. Structurally, a CAR is composed of an extracellular domain which includes an antigen-binding domain typically taken from the single-chain variable fragment (scFv) of an antibody, a transmembrane domain which traverses the cell membrane, costimulatory domains needed to enhance CAR T cell functions such as OX40, 4-1BB or CD28, and an intracellular domain comprised a T cell signaling domain (CD3-zeta) (see Figure 1A) [2]. CAR T cells targeting CD19 or B cell maturation antigen (BCMA) are FDA approved for the treatment of relapsed/refractory B cell lymphoma/leukemia and multiple myeloma, respectively, following high rates of complete responses during clinical trials [3,4,5,6,7,8,9,10]. Despite these successes against hematological malignancies, CAR T cell therapy has remained relatively ineffective against non-hematological solid tumors for numerous reasons including lack of persistence, exhaustion, lack of infiltration into the immunosuppressive tumor microenvironment (TME), and decreased antigen expression by tumor cells [11].

Targeted radionuclide therapy (TRT), also referred to as radiopharmaceutical therapy, selectively delivers radioactive compounds to malignant cells using tumor-homing ligands [12,13,14]. TRT enables the systemic and relatively selective delivery of radiation to all sites of disease in patients with metastatic disease and has demonstrated a survival benefit in patients with midgut neuroendocrine tumors and castration-resistant prostate cancer [15,16]. Furthermore, emerging data suggest that low-dose radiation delivered by TRT elicits a favorable immune response by activating and enhancing the infiltration of endogenous T cells into the TME [17,18]. Thus, we hypothesize that TRT may help overcome some of the shortcomings of CAR T cell therapy against solid tumors by inducing a pro-inflammatory TME. However, prior to testing this hypothesis in vivo, numerous questions must be addressed. Of particular importance is the type of radionuclide, alpha (α) versus beta (β) particle emitters, and how the timing and sequence of TRT relative to CAR T cell administration can impact the therapeutic efficacy, such that tumor susceptibility to CAR T cells can be augmented while not abrogating CAR T cell function.

Herein, we evaluated in vitro the impact of the α-particle emitter actinium-225 (^225^Ac) and the β-particle emitter lutetium-177 (^177^Lu) on the viability, cytotoxic function, and phenotype of third-generation anti-GD2 CAR T cells. These third-generation anti-GD2 CAR T cells were produced using a virus-free CRISPR-based approach, with the CAR construct (Figure 1B) knocked into the T cell receptor alpha constant gene (*TRAC*) locus, resulting in anti-GD2 CAR T cells devoid of their endogenous T cell receptors (Figure 1C) [19].

## 2. Materials and Methods

### 2.1. Cell Lines

The GD2-expressing CHLA-20 cell line (human neuroblastoma) and M21 cell line (human melanoma) were kindly donated by Dr. Mario Otto and Dr. Ralph Reisfeld, respectively, and grown in Dulbecco’s Modified Eagle Medium (DMEM) with high glucose (ThermoFisher Scientific, Waltham, MA, USA) supplemented with 10% fetal bovine serum (FBS) (Avantor, Radnor Township, PA, USA) and 1% penicillin–streptomycin (ThermoFisher Scientific, Waltham, MA, USA). These cells were maintained at 37 °C in 5% CO_2_. Cell line authentication was completed using genomic short-tandem repeat analysis (Idexx BioAnalytics, Columbia, MO, USA) and by cell morphology per ATCC guidelines. *Mycoplasma* testing was performed on a regular basis to rule out contamination using the Mycoplasma Detection Kit MycoStrip™ (InvivoGen, San Diego, CA, USA).

### 2.2. Anti-GD2 CAR T Cells

The CAR transgene from a pSFG.iCasp9.2A.14G2A-CD28-OX40-CD3ζ retroviral CAR plasmid (gift from Dr. Malcom Brenner, Baylor College of Medicine, Houston, TX, USA) was used to generate virus-free CRISPR anti-GD2 CAR T cells as previously described [19]. Primary human T cells were isolated from peripheral blood mononuclear cells from healthy donors using an Institutional Review Board-approved protocol from the University of Wisconsin-Madison (#2018-0103, Madison, WI, USA). The anti-GD2 CAR T cells were cultured in ImmunoCult-XF T cell Expansion Medium (Stemcell™ Technologies, Vancouver, BC, Canada) supplemented with 500 U/mL of IL-2 (Peprotech, Cranbury, NJ, USA) and maintained 37 °C in 5% CO_2_.

### 2.3. In Vitro Dosimetry

All in vitro dosimetry studies were performed in 6-well cell culture plates. Serial dilutions of various activity of ^177^Lu or ^225^Ac were made in the cell culture medium and added to each well. To determine the amount of activity needed to deliver a desired absorbed dose, Monte Carlo simulations were performed in representative geometries as previously reported [20]. The mean absorbed dose to cells was calculated using an extension of RAPID, which is a radiopharmaceutical therapy dosimetry platform that runs on the Geant4 Monte Carlo toolkit [21]. A model of a flat-bottom 6-well plate was developed in RAPID using manufacturing specifications where the diameter and height of each well were, respectively, 36 mm and 10.7 mm. The cell volume can be defined as a thin water-equivalent layer at the bottom of the well. To confirm the validity of these Monte Carlo dosimetry calculations, absorbed doses in the wells were also measured using thermoluminescent dosimeters (TLDs) placed at the bottom of wells containing cell culture medium with various activities of β particle emitter ^90^Y. The TLDs were harvested after 1 half-life and analyzed by the University of Wisconsin-Madison Radiation Calibration Laboratory (Calibration Cert # 1664.01). A calibration curve was obtained and used to determine the activity needed to deliver a given absorbed dose to the cells.

### 2.4. In Vitro Anti-GD2 CAR T Cell Irradiation

Anti-GD2 CAR T cells (10^6^ per well) were irradiated with various doses of radiation delivered over 3 days by ^177^Lu or ^225^Ac diluted in 3 mL of ImmunoCult-XF T cell Expansion Medium per well on a 6-well plate. Doses of radiation delivered by ^177^Lu were 1, 2 and 6 Gy, corresponding to activity concentrations of 13.1, 26.3 and 78.7 μCi/mL, respectively. With ^225^Ac, 1 and 2 Gy were delivered, corresponding to activity concentrations of 0.088 and 0.176 μCi/mL, respectively.

### 2.5. In Vitro Anti-GD2 CAR T Cells and Tumor Cells Co-Culture

After the anti-GD2 CAR T cells were irradiated, they were harvested and washed 3 times with phosphate buffer saline (PBS) to remove residual radionuclide. A subsequent trypan blue viability assay was performed on the irradiated anti-GD2 CAR T cells, and viable CAR T cells were co-cultured with tumor cells with an effector to target (E:T) ratio of 10:1 (100,000 anti-GD2 CAR T cells to 10,000 tumor cells) in 100 µL of DMEM in a 96-well for 24 h.

### 2.6. Flow Cytometric Analysis

Cells were harvested, washed with PBS, and resuspended into single-cell solution in PBS as previously reported [22]. Fc blocking (Biolegend, San Diego, CA, USA) and Live/Dead staining with Ghost Red Dye 780 (Tonbo Biosciences, San Diego, CA, USA) were performed for 10 min at 4 °C. The fluorophore-conjugated antibodies including anti-CD45-APC (Biolegend), anti-GD2-PE-Dazzle584 (Biolegend), anti-PD-1-PE (Biolegend), anti-CD69-BV510 (Biolegend), and anti-NKG2D-BV605 (Biolegend) were incubated for 20 min at 4 °C and washed with 2% FBS in PBS. The analysis of the sample was performed using the Attune NxT Flow Cytometer (ThermoFisher Scientific, Waltham, MA, USA), and the collected data were analyzed using FlowJo software v10.

### 2.7. Statistical Analysis

The anti-GD2 CAR T cells’ viability, cytotoxic activity, and marker expression were summarized for each group in terms of means (triplicate measurements) and standard deviations. The effect of ^225^Ac and ^177^Lu on CAR T cells’ viability, cytotoxic activity, and marker expression were analyzed using one-way analysis of variance (ANOVA). Tukey’s Honestly Significant Difference (HSD) method was used to control the type I error when conducting multiple pairwise comparisons between the dose level group and the no radiation control group. Statistical analyses were performed using Prism 9 (GraphPad) software. All reported *p*-values were two-sided, and *p* < 0.05 was used to define statistical significance.

## 3. Results

### 3.1. Radiation Delivered by ^177^Lu and ^225^Ac Results in a Dose-Dependent Anti-GD2 CAR T Cell Death

Combining in vivo TRT and CAR T cell therapy will undeniably lead to the exposure of CAR T cells to radiation, which could be detrimental to their viability. As such, to evaluate the effect of radiation delivered by TRT on CAR T cells, third-generation anti-GD2 CAR T cells were exposed to various doses of radiation delivered by ^177^Lu or ^225^Ac. Six-well plates containing anti-GD2 CAR T cells were incubated in media containing the specific activity of ^177^Lu or ^225^Ac needed to deliver radiation doses of 0, 1, 2 or 6 Gy by day 3. Following the delivery of the radiation, the anti-GD2 CAR T cells were harvested, and their viability was evaluated by flow cytometry (Figure 2A) using live/dead staining of the CD45+ cells (Figure 2B). A dose-dependent anti-GD2 CAR T cell death was observed with both radionuclides. However, 1 and 2 Gy of ^177^Lu appeared to be less cytotoxic than comparable doses of ^225^Ac. Compared to the non-irradiated anti-GD2 CAR T cells, 1 and 2 Gy of radiation delivered by ^225^Ac reduced the viability of anti-GD2 CAR T cells to 34% and 5%, respectively (Figure 2B). In contrast, after the delivery of 1, 2 and 6 Gy of radiation by ^177^Lu, the viability of anti-GD2 CAR T cells decreased to 87%, 72% and 53%, respectively, compared to non-irradiated CAR T cells (Figure 2B).

### 3.2. Radiation Delivered by ^177^Lu and ^225^Ac Enhances the Cytotoxic Activity of Anti-GD2 CAR T Cells against the GD2-Expressing Neuroblastoma Cell Line CHLA-20

In addition to evaluating the effect of radionuclide-delivered radiation on the viability of the anti-GD2 CAR T cells, we were also interested in determining the impact of such radiation on the effector function of the anti-GD2 CAR T cells without exposing the tumor cells to radiation. To this end, following irradiation, the anti-GD2 CAR T cells were washed and then co-cultured with the GD2-expressing neuroblastoma cell line CHLA-20 for 24 h. The killing potential of the irradiated anti-GD2 CAR T cells was determined by measuring the viability of CHLA-20 cells by flow cytometry (Figure 3A). Viable CHLA-20 cells were identified as CD45 negative and live/dead stain (Ghost red) negative (CD45-/Ghost red). As expected, non-irradiated anti-GD2 CAR T cells displayed a potent cytotoxicity activity against CHLA-20 cells. The viability of the CHLA-20 cells decreased to 7.0% after 24 h co-culture with non-irradiated anti-GD2 CAR T cells compared to 52% in the absence of anti-GD2 CAR T cells (Figure 3B). The irradiation of anti-GD2 CAR T cells prior to co-culture with CHLA-20 cells significantly enhanced their cytotoxic activity of the CAR T cells, resulting in a near complete eradication of CHLA-20 cells, as evidenced by tumor viabilities ranging from 1.3 to 1.6% (Figure 3B). For the type and doses of radionuclides used in this study, the enhanced cytotoxicity of the anti-GD2 CAR T cell was similar for all conditions tested (Figure 3B). The radiation-induced enhancement of anti-GD2 CAR T cell cytotoxic activity was also observed when these same CAR T cells were tested against the human melanoma cell line M21, which also expresses GD2 (Appendix A).

### 3.3. Radiation Delivered by ^177^Lu and ^225^Ac Does Not Impact the Expression of Exhaustion and Activation Markers on Anti-GD2 CAR T Cells

To further characterize the effect of ^225^Ac or ^177^Lu-delivered radiation on anti-GD2 CAR T cells and understand the underlying mechanism of the radiation-induced cytotoxic enhancement of anti-GD2 CAR T cells, the expression of the T cell exhaustion marker PD-1, activation marker CD69, and activating receptor NKG2D were evaluated by flow cytometry. As previously described, following the irradiation of the anti-GD2 CAR T cells with the various radiation doses of ^225^Ac or ^177^Lu, the anti-GD2 CAR T cells were harvested, and the expression of cell surface markers was determined by flow cytometry (Figure 4A). Compared to non-irradiated anti-GD2 CAR T cells, 1 Gy delivered by ^225^Ac resulted in a slight increase in expression of PD-1 that was not significant. In contrast, radiation (1, 2, 6 Gy) delivered by ^177^Lu resulted in the decrease in PD-1 expression, which was also not significant (Figure 4B). CD69 and NKG2D expressions were increased following the delivery of radiation by both radionuclides, albeit this was non-significant (Figure 4C,D).

## 4. Discussion

Due to the increasing therapeutic role of TRT in the clinical management of cancers and the immunostimulatory effects at low-dose TRT, the combination of TRT with CAR T cell therapy may represent a therapeutic approach for selected patients with metastatic solid tumors [17,20]. As such, we investigated the in vitro effects of TRT on CAR T cells’ viability and potency to evaluate the influence of different doses of ^225^Ac and ^177^Lu (α vs. β-particle emitters). We tested whether selected doses of radiation delivered by these radionuclides can enhance the effector function of CAR T cells while minimizing the deleterious effects on CAR T cell viability.

The dose-dependent death of anti-GD2 CAR T cells observed in response to radiation delivered by ^177^Lu and ^225^Ac is expected because radiation is cytotoxic to lymphocytes, especially CD8+ T cells from which anti-GD2 CAR T cells are manufactured [23]. Although such dose-dependent death was observed with radiation delivered by either radionuclide, the biological effect of a similar dose of radiation (1 Gy) delivered by the α-emitter ^225^Ac was higher than that of the same dose delivered by the β-emitter ^177^Lu. This difference in biological effect is likely attributable to the linear energy transfer (LET) difference between these two radionuclides, which has been previously reported [24]. Despite having a shorter range in tissue (0.05–0.08 mm) compared to β-emitters (1–5 mm), α-emitters, due to their higher LET (50–230 keV/μm), induce 10–20 double-strand DNA breaks (DSBs) per 10 μm, resulting in potent cytotoxicity, whereas β-emitters induce more single-strand DNA breaks, which are easily repaired [25,26,27]. α-particle emitters induce multiple DSBs that are in close vicinity, resulting in the depletion of p53-binding protein 1 (53BP1), which is a DNA damage response protein required for double-strand DNA breaks repair. Such depletion leads to an insufficient amount of 53BP1 to adequately repair all DSBs, further accentuating the biological effectiveness of high LET α-particle emitters [28].

Using a prostate cancer in vitro model, the in vitro therapeutic efficacy of ^225^Ac-PSMA was found to be 4.2 times that of ^177^Lu-PSMA, and such a difference was solely attributed to the higher relative biological effectiveness (RBE) of ^225^Ac compared to ^177^Lu [29]. However, in this study, the in vitro CAR T cell death induced by 1 Gy of ^225^Ac is only 2.2 times that of ^177^Lu (Figure 2). This discrepancy could be attributed to the different in vitro models used in this study, the method used to access the therapeutic efficacy and/or to the fact that our in vitro dosimetry enabled the comparison of a similar dose of radiation (Gy) which mediates the biological effects delivered by a radionuclide rather than comparing activity. It is also possible that the mechanism of radionuclide-induced cell death of T cells is different from that of tumor cells.

Although radiation delivered by external beam radiation therapy (EBRT) upregulates the expression of PD-1 on CD8+ T cells, which can be detrimental to CAR T cell function, surprisingly, we did not observe such upregulation with ^177^Lu or ^225^Ac in the conditions tested [30]. This may be due to the inherent design of this third-generation anti-GD2 CAR with the CRISPR knock-in of the CAR construct into the *TRAC* locus, yielding CAR T cells devoid of T cell receptors, and it may represent another therapeutic benefit of combining CAR T cell therapy with TRT instead of EBRT [19].

The lack of significant impact of TRT on the expression of the activating receptor NKG2D appears to be in contradiction to the findings by others who demonstrated that low-dose radiation (1 and 3 Gy) delivered by EBRT induces an upregulation of the NKG2D receptors on T and NK cells [30]. Such a discrepancy could be attributed to the difference in the form of radiation delivered (EBRT vs. TRT) and/or in the method used to detect the NKG2D receptor protein expression. EBRT was previously shown to upregulate NKG2D expression by quantifying mRNA expression, whereas in this study, NKG2D receptor expression was quantified at the protein level by flow cytometry [31]. While the increase in mRNA expression may be used to suggest an upregulation of a gene of interest, this may not necessarily translate into a change in protein expression in the proper cellular compartment such as lysosome, mitochondrial or cell surface [32,33].

Radiation delivered by EBRT enhances the cytotoxic activity of CAR T cells including against antigen-negative malignant cells [34]. However, prior to this study, such enhancement has not been demonstrated with TRT. While ^177^Lu or ^225^Ac did not significantly decrease the expression of PD-1 or increase the expression of the NKG2D receptor or activation marker CD69, it is possible that the presence of the TRT in the milieu may have selected for more radioresistant CAR T cells subclones with potent cytolytic activity. It is also plausible that the low-dose radiation delivered by ^177^Lu or ^225^Ac stimulated ATP production (mitochondrial excitation), which can boost the anti-oxidative capacity of CAR T cells and enhance their radiosensitivity and survival [35].

Based on our in vitro studies, the low dose of radiation tested herein (1 or 2 Gy) delivered by α-emitters might not be ideal for a potential therapeutic combination with CAR T cells. Instead, β-emitters may be the preferred candidates, given that the 1 Gy delivered by a β-particle emitter enhanced CAR T cell cytotoxic function to the same extent as all other radiation doses evaluated (2 or 6 Gy) while inducing lower rates of anti-GD2 CAR T cell death compared to those higher doses. These observations suggest that with regard to the sequencing between TRT and CAR T cells combination, the infusion of CAR T cells may optimally be performed after the administration of TRT to minimize the exposure of CAR T cells to radiation while enabling the activation of inflammatory effects in the TME and phenotypic effects on tumor cells that maximize the sensitivity to CAR T cells killing, which are activated optimally at higher doses. While our in vitro study suggests that 1 Gy or lower delivered by the β-emitter ^177^Lu might be well tolerated by CAR T cells, the higher tested doses of 2 or 6 Gy may be considered if there is an adequate delay between the TRT delivery and CAR T cell infusion. This delay should enable the radioactive decay of the radionuclide such that when the CAR T cells are infused, the remaining activity of the radionuclide in the TME may only deliver a radiation dose of 1 Gy or less. In vivo testing is sorely needed. We speculate that using a radiation dose higher than 1 Gy may be an attractive option, as it will enable the tumoricidal effect of TRT on malignant cells to be harnessed, subsequently reducing the tumor volume against which CAR T cells have to mount an effective therapeutic response.

Aside from the significantly higher cytotoxicity of the α-emitter ^225^Ac on the CAR T cell compared to the β-emitter ^177^Lu, additional factors support the preference of ^177^Lu over ^225^Ac. To achieve the adequate delivery of the radionuclide in vivo, the incorporation of a radiometal chelating moiety in the pharmacophoric model of the TRT is needed. To date, 1,4,7,10-tetraazacyclododecane-1,4,7,10-tetraacetic acid (DOTA) is the commonly used chelator [36]. However, the stable retention of ^225^Ac and its daughters by this chelator has been challenging, resulting in it a non-specific release of the radionuclide from the TRT targeting vector and thus toxicities [37]. Furthermore, because of the relatively longer range in tissue of ^177^Lu compared to ^225^Ac, ^177^Lu may be advantageous in irradiating uniformly heterogeneous tumors in vivo [24].

Various limitations can be highlighted in this study, including the in vitro character of the study, the evaluation of only a select number of exhaustion or activation markers, the relatively limited numbers of radionuclide and doses evaluated. Nevertheless, this study provides critical insights on key questions regarding the optimal doses of radiation, type of radionuclide, and timing and sequence of TRT and CAR T cell combinations to enable the rational design of in vivo studies evaluating the combination of CAR T cell therapy and TRT in solid tumor models.

## 5. Conclusions

While the immunostimulatory effects of both α and β-emitter radionuclides have been reported and shown to potentiate immune checkpoint inhibitors, they also have the potential to be useful in overcoming some of the shortcomings of CAR T cell therapy against solid tumors. However, proper selection of the radionuclide is critical for in vivo studies to efficiently harness the immunostimulatory effect while minimizing the deleterious effects on CAR T cell viability and potency. Such selection should be guided by the properties of the radionuclides including LET, half-life, and range in tissue and should lead to establishing an optimal sequencing and timing between the delivery of TRT and CAR T cells. Lastly, because TRT selectively delivers radiation to all sites of disease, it may enable the immunostimulatory effect of radiation in combination with CAR T cells to be exploited in patients with widely metastatic solid tumors in whom treatment-related toxicities render EBRT to all sites of disease infeasible.

## Figures and Tables

**Figure 1 cancers-16-00191-f001:**
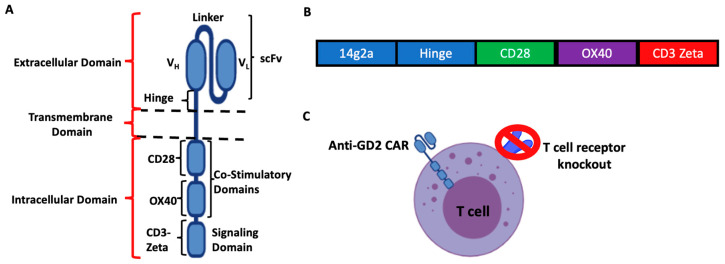
Third-generation virus-free CRISPR anti-GD2 CAR T cells. (**A**) Schematic of the domains of the third generation CAR. The extracellular domain includes the scFV (V_H_ and V_L_ chains connected by a linker) and a hinge. The intracellular domain is comprised of 2 costimulatory domains (CD28 and OX40) and a signaling domain (CD3-Zeta). These 2 domains are connected by a transmembrane domain (CD28 transmembrane domain). (**B**) Schematic of the third generation anti-GD2 CAR construct inserted into the human T cell receptor alpha constant gene (*TRAC*). (**C**) Schematic of the anti-GD2 CAR T cell used experimentally. It expresses the anti-GD2 CAR but is devoid of T cell receptor due to the CRISPR knockout of the human *TRAC* gene. scFv: single-chain variable fragment; V_H_: heavy chain; V_L_: light chain.

**Figure 2 cancers-16-00191-f002:**
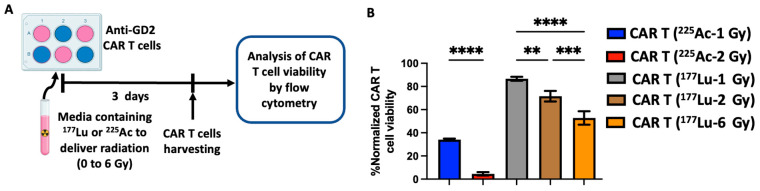
Dose-dependent effect of ^225^Ac and ^177^Lu on the viability of anti-GD2 CAR T cells. (**A**) Experimental scheme: anti-GD2 CAR T cells were incubated in cell culture medium containing free ^225^Ac or ^177^Lu with activities calculated to deliver a radiation dose between 1 and 6 Gy by day 3. The CAR T cells were harvested, and their viability was analyzed by flow cytometry using a live/dead staining. Viable CAR T cells were determined as CD45+ and live/Dead-Ghost red-. (**B**) Both ^225^Ac and ^177^Lu led to a dose-dependent CAR T cell death. One-way ANOVA with Tukey’s multiple comparisons correction ** *p* = 0.0024; *** *p* = 0.0001; **** *p* < 0.0001. Each bar represents the mean (triplicate measurements) and error bar represents the standard deviation.

**Figure 3 cancers-16-00191-f003:**
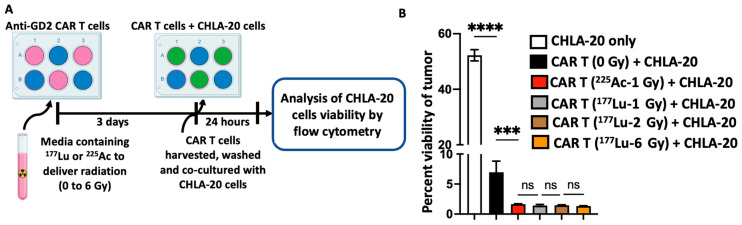
Dose-independent effect of ^225^Ac and ^177^Lu on anti-GD2 CAR T cell cytotoxicity. (**A**) Experimental scheme: After irradiation of CAR T cells by ^225^Ac or ^177^Lu, the CAR T cells were harvested, washed and trypan blue viability assay was performed. The viable CAR T cells after irradiation were co-cultured with the GD2-expressing human neuroblastoma cell line CHLA-20 for 24 h at an E:T ratio of 10:1 (viable anti-GD2 CAR T cells: tumor cells). (**B**) The viability of the CHLA-20 cells was analyzed by flow cytometry. The exposure of anti-GD2 CAR T cells to radiation delivered by radionuclide enhances their cytotoxicity against CHLA-20 cells to a similar degree for all doses tested for ^225^Ac or ^177^Lu. One-way ANOVA with Tukey’s multiple comparisons correction *** *p* < 0.001; **** *p* < 0.0001; ns: not significant. Each bar represents the mean (triplicate measurements), and the error bar represents the standard deviation. The cytotoxic activity of CAR T cells irradiated with 2 Gy of radiation delivered by ^225^Ac was not performed because of very low CAR T cell viability after irradiation.

**Figure 4 cancers-16-00191-f004:**
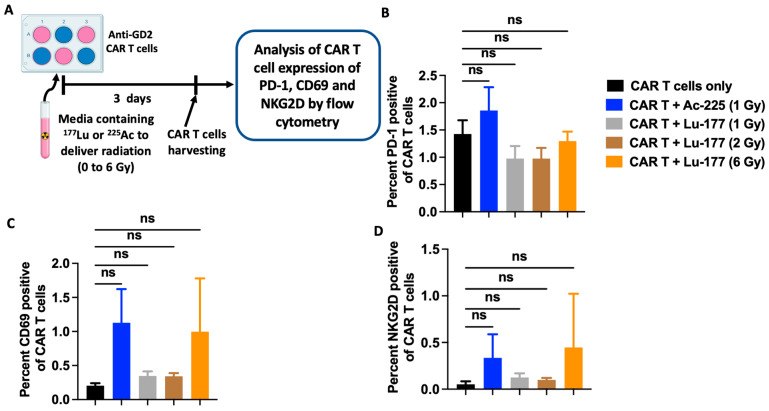
^225^Ac or ^177^Lu does not impact the expression of exhaustion and activation markers on anti-GD2 CAR T cells. (**A**) Experimental scheme: anti-GD2 CAR T cells were incubated in cell culture medium containing ^225^Ac or ^177^Lu with activities calculated to deliver a radiation dose between 1 and 6 Gy by day 3. CAR T cells were harvested, and the expression of exhaustion and activation markers was analyzed by flow cytometry. (**B**) The expression of the exhaustion marker PD-1 does not change following irradiation by ^225^Ac or ^177^Lu. (**C**) Similarly, no major impact is seen on the expression of the T cell activation marker CD69. (**D**) Irradiation does not result in a major impact on the activation marker NKG2D. One-way ANOVA with Tukey’s multiple comparisons correction; ns: not significant. Each bar represents the mean (triplicate measurements), and the error bar represents the standard deviation. The effects of 2 Gy irradiation delivered by ^225^Ac on the receptors’ expression was not evaluated because of very low CAR T cell viability after irradiation. ns: not significant.

## Data Availability

The data presented in this study are available upon request to the corresponding author.

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
