# Peer review of "Comparative Study of the Effect of Radiation Delivered by Lutetium-177 or Actinium-225 on Anti-GD2 Chimeric Antigen Receptor T Cell Viability and Functions"

_cancers, 2023, doi:10.3390/cancers16010191_

Round 1
Reviewer 1 Report
Comments and Suggestions for Authors
This study explores the combination of chimeric antigen receptor (CAR) T cells with targeted radionuclide therapy (TRT) using 177Lu and 225Ac against solid tumors. Both radionuclides demonstrated dose-dependent toxicity on CAR T cells but enhanced their cytotoxic activity against neuroblastoma and melanoma cells. Although 177Lu is favored due to better CAR T cell viability, both radionuclides show promise for overcoming CAR T therapy limitations in solid tumors. The study emphasizes the crucial role of radionuclide selection based on properties such as linear energy transfer and half-life for optimal therapeutic synergy while minimizing adverse effects.
This study is pivotal as it investigates the novel combination of chimeric antigen receptor (CAR) T cells with targeted radionuclide therapy (TRT) against solid tumors. Addressing CAR T therapy limitations, it highlights the potential synergistic benefits and underscores the critical role of radionuclide selection for optimizing therapeutic efficacy in treating widely metastatic solid tumors. However, the present form of the manuscript required modification before taking any positive decision.
1. What is the main mechanism underlying the enhanced cytotoxic activity of CAR T cells against neuroblastoma and melanoma cells following irradiation with both 177Lu and 225Ac?
2. Considering the dose-dependent toxicity, how to determine the optimal radiation dose for achieving an effective therapeutic balance between CAR T cell viability and enhanced cytotoxicity?
3. specific considerations and properties of 177Lu make it more favorable over 225Ac for in vivo applications against solid tumors. Explain in the manuscript.
4. It is advised to thoroughly review and rectify all inaccuracies and grammatical mistakes to enhance the overall quality of the paper.
Comments on the Quality of English Language
It is advised to thoroughly review and rectify all inaccuracies and grammatical mistakes to enhance the overall quality of the paper.
Author Response
- What is the main mechanism underlying the enhanced cytotoxic activity of CAR T cells against neuroblastoma and melanoma cells following irradiation with both 177Lu and 225Ac?
Response: We thank the reviewer for this thoughtful question. To elucidate this mechanism, we evaluated whether the irradiation of the CAR T cells decreased their expression of the immune checkpoint PD-1 which could result in enhancing their cytotoxic activity. Alternatively, the upregulation of the activating receptor NKG2D could also result in an enhancement of the CAR T cell cytotoxicity. However, there was no statistical significance in the expression of the aforementioned receptors with or without radiation. As such, we are hypothesizing that the low dose radiation may be stimulating ATP production which can enhance the radioresistance of the CAR T cells as previously described by Hietanen et al. Anticancer Research 35, 5193-5200 (2015)
- Considering the dose-dependent toxicity, how to determine the optimal radiation dose for achieving an effective therapeutic balance between CAR T cell viability and enhanced cytotoxicity?
Response: Thank you for bringing up this question. Based on our CAR T cell viability, 177Lu may be the optimal radionuclide for in vivo experiments to evaluate the combination of TRT and CAR T cell therapy. Using the 177Lu, it also follows that it is best to limit the radiation dose received by CAR T cells to £ 1Gy since compared to non-irradiated CAR T cells, 1 Gy of radiation delivered by 177Lu only reduces CAR T cell viability by only 14%. To minimize irradiation of CAR T cells in vivo, TRT should be administered first followed by waiting period that can allow the radionuclide to decay, thus reducing the amount of residual radiation by the time CAR T cells are infused. For example, treating first with 2 Gy of 177Lu then waiting for 1 half-life (6.6 days for 177Lu) will make sure that by the time the CAR T cells are infused, they will be exposed to a residual radiation of only 1 Gy and the cytotoxic activity enhancement ensuing from the exposure to 1 Gy would have been similar to that observed with 2 or 6 Gy.
- Specific considerations and properties of 177Lu make it more favorable over 225Ac for in vivo applications against solid tumors. Explain in the manuscript.
Response: We have now included in the revised manuscript in the discussion section a paragraph highlighting in addition to the in vitro data we present some of the benefits of using 177Lu over 225Ac for in vivo applications against solid tumors.
- It is advised to thoroughly review and rectify all inaccuracies and grammatical mistakes to enhance the overall quality of the paper.
Response: We thank the reviewer for bringing this to our attention. We have subsequently reviewed the manuscript again to address any inaccuracies and grammatical errors that we may have inadvertently missed.
Reviewer 2 Report
Comments and Suggestions for Authors
The paper "Comparative Study of the Effect of Radiation Delivered by Lutetium-177 or Actinium-225 on Anti-GD2 Chimeric Antigen Receptor T Cell Viability and Functions" by Dr. Sodji et al. explores an effect where irradiated T cells have been found to have enhanced effect on tumors. Dr. Sodji et al. explores this effect on chimeric antigen receptor (CAR) T cells, investigating both CAR T cell survival and the effect on human cancer cells in vitro. Irradiation is performed with two nuclides used for targeted radionuclide therapy (TRT), potentially opening the possibility of CAR T cell irradiation in vivo from the TRT in the patient. From these in vivo experiments, the authors find an enhanced cytotoxic effect by the irradiated CAR T cells on cancer cells.
While the enhancing effect by radiation on the tumor microenvironment (TME) has been explored before, at cited by the article [17,18], the authors appear to be first to study the effect and possibilities in combination with CAR T cells. Their article is interesting and to the point. In some places, however, it can be clarified, see comments below.
MINOR
1. A general note on notation: Therapy involving radionuclides like Lu-177 or Ac-225 will in most cases involve more than just the radionuclide - the radionuclide will be part of a larger molecule. To reflect this, the term "radiopharmaceutical therapy (RPT)" is increasingly being used. Citing from ICRP Publication 140, Radiological Protection in Therapy with Radiopharmaceuticals: "Therapy with radiopharmaceuticals is referred to by many synonymous terms, including ‘targeted radionuclide therapy’, ‘unsealed source therapy’, ‘systemic radiation therapy’, and ‘molecular radiotherapy’. In this publication, the generic term ‘radiopharmaceutical therapy’ is used for consistency with other ICRP and ICRU publications." (Footnote on page 13 of IRCP 140). It is suggested (but not required) that the authors considers using the term RPT instead of TRT. This reviewer has noted, that the term is in fact already used in the paper in the subsection "In vitro dosimetry". (To avoid confusion, this review will use TRT when the paper in its present form writes TRT.)
2. In the abstract, it was at first confusing to read that "(TRT) can deliver low dose of radiation". The general aim in TRT is to deliver a high, but focused dose! Your paper is not about this general aim, but to make the text clearer, it is suggested to set the scene by mentioning the immunostimulatory effects before TRT. For instance: "Earlier studies have shown general immunostimulatory effects of low-dose radiation. A source of low-dose radiation can be the radioactivity from targeted radionuclide therapy (TRT), but this has never been combined with CAR T cells ..."
3. The closing paragraph of the Introduction: It takes some work of the reader to see structure of the sentence: "Herein, we evaluated in vitro the impact of actinium-225 (225Ac), an α-particle emitter, and lutetium-177 (177Lu), a β-particle emitter on the viability, cytotoxic function, and phenotype of third generation anti-GD2 CAR T cells." Suggested rephrasing: "Herein, we evaluated in vitro the impact of the α-particle emitter actinium-225 (225Ac) and the β-particle emitter lutetium-177 (177Lu) on the viability, cytotoxic function, and phenotype of third generation anti-GD2 CAR T cells." Or split in two sentences, for instance like this: "Herein, we evaluated in vitro the impact of radiation dose on the viability, cytotoxic function, and phenotype of third generation anti-GD2 CAR T cells. The radiation dose was delivered by actinium-225 (225Ac), an α-particle emitter, or by lutetium-177 (177Lu), a β-particle emitter."
4. Only the radiation doses (1, 2, or 6 Gy) are reported in the paper, not the activities (Bq). It will be of interest to the reader also to know what activities are used. This includes the use of 90Y briefly mentioned in the subsection "In vitro dosimetry".
5. The irradiation is performed on a 6-well plate. While the wells are physically separated, they will not be completely separated regarding radiation. Cross-irradiation among wells is likely a minor or very minor effect, but please address it, for instance by explaining why the effect is so small it can be ignored, if this is the case.
6. Figures 1, 2, and 3 use very small print. It is suggested to enlarge the figures to make the font size comparable to that of Figure 4.
7. Figure 3 appears to be missing a column for 2Gy-irradiation using 225Ac (Fig. 2 has Ac before Lu, Fig. 3 has Lu before Ac). Furthermore, is it by intention that the order and notation differs slightly from Figure 2? (Fig 2 writes "225Ac (1 Gy)", Fig. 3 writes "225Ac-1Gy")
8. By the end of the Results section, non-significant "trends" in Figure 4 are described. Please avoid describing "trends" when the results are not significant. In fact, the text on "trends" is counter to the legend of Figure 4, which opens with the words: "225Ac or 177Lu DOES NOT IMPACT the expression ..."
9. In the Discussion, the difference between alpha- and beta-irradiation is discussed. It is well-known that alpha particles have a higher biological effectiveness than beta particles (and photons), but the present formulations (page 7) can be read as if this was a new finding: "This difference is biological effect is likely attributable to the linear energy transfer (LET) difference between these two radionuclides." Please make it more clear that the difference is not a new result, e.g. by including a reference to a work on relative biological effectiveness. (If in doubt where to find a good reference, take a look at ICRP Publication 92, which can be freely downloaded from https://icrp.org/page.asp?id=5 ).
10. On page 7, the meaning of this phrase is unclear: "... our in vitro dosimetry enabled the comparison of a similar dose of radiation delivered by a radionuclide instead of activity." It is not clear how "radionuclide" opposes "activity" ("radionuclide instead of activity"). Is the meaning that dosimetry enabled comparison with the actual radiation dose (Gy) rather than with the activity (Bq) used? Please rephrase.
11. The authors suggest that their findings may be used to combined CAR T cell therapy with TRT, to irradiate the CAR T cells in vivo by the activity from the radioactivity given as TRT. Surely an idea worth further studies. Have the authors considered the possibility of in vitro irradiation of CAR T cells to stimulate the CAR T cells before application of the CAR T cells to the patient? This might broaden the application to patients not given TRT, although it still includes the handling of radioactivity by the staff. If a similar effect could be obtained with photon irradiation of the CAR T cells, it might be performed with easy-accessible radionuclides like 99mTc, or even without radioactivity, using an x-ray device. These are just ideas for thought.
VARIOUS / VERY MINOR
12. In the first paragraph of the introduction, part of a sentence seems to be missing: "Structurally, a CAR is composed of ... and (Fig. 1A)." - and what? (co-stimulatory domains?)
13. Later in the Introduction: "Thus, we postulated that TRT may help ..." Is "postulated" = claimed that it was so, the right word here? Suggestion: "Thus we propose that TRT may help ..." (the change from past to present tense in this place is part of the suggestion - for the reader, past tense would naturally refer to earlier work.)
14. Bottom of page 2: There seems to be unwanted repetition in the phrase "... and how the timing the timing ..."
15. Middle of page 5: It is suggested to rephrase the sentence "The viability ... decreased to 6.96% ... compared to 52.2%." Suggestion: "The viability ... decreased from 52% to 7.0% ..." (the suggested fewer significant figures are based on the error bars in Figure 3.)
16. The abbreviation "PBS" is used but not defined.
17. Font/formatting: In a few places, the text font deviates from the standard. This seems to be in parts of the text where the Greek letters α and β are used.
Author Response
- A general note on notation: Therapy involving radionuclides like Lu-177 or Ac-225 will in most cases involve more than just the radionuclide - the radionuclide will be part of a larger molecule. To reflect this, the term "radiopharmaceutical therapy (RPT)" is increasingly being used. Citing from ICRP Publication 140, Radiological Protection in Therapy with Radiopharmaceuticals: "Therapy with radiopharmaceuticals is referred to by many synonymous terms, including ‘targeted radionuclide therapy’, ‘unsealed source therapy’, ‘systemic radiation therapy’, and ‘molecular radiotherapy’. In this publication, the generic term ‘radiopharmaceutical therapy’ is used for consistency with other ICRP and ICRU publications." (Footnote on page 13 of IRCP 140). It is suggested (but not required) that the authors consider using the term RPT instead of TRT. This reviewer has noted, that the term is in fact already used in the paper in the subsection "In vitro dosimetry". (To avoid confusion, this review will use TRT when the paper in its present form writes TRT.)
Response: We thank the reviewer for this suggestion. Instead of using radiopharmaceutical therapy only, we included both term targeted radionuclide therapy and radiopharmaceutical therapy in the introduction section to allow the reader to understand that both are synonymous.
- In the abstract, it was at first confusing to read that "(TRT) can deliver low dose of radiation". The general aim in TRT is to deliver a high, but focused dose! Your paper is not about this general aim, but to make the text clearer, it is suggested to set the scene by mentioning the immunostimulatory effects before TRT. For instance: "Earlier studies have shown general immunostimulatory effects of low-dose radiation. A source of low-dose radiation can be the radioactivity from targeted radionuclide therapy (TRT), but this has never been combined with CAR T cells ..."
Response: We have clarified this as suggested by the reviewer. Please see the abstract.
- The closing paragraph of the Introduction: It takes some work of the reader to see structure of the sentence: "Herein, we evaluated in vitrothe impact of actinium-225 (225Ac), an α-particle emitter, and lutetium-177 (177Lu), a β-particle emitter on the viability, cytotoxic function, and phenotype of third generation anti-GD2 CAR T cells." Suggested rephrasing: "Herein, we evaluated in vitrothe impact of the α-particle emitter actinium-225 (225Ac) and the β-particle emitter lutetium-177 (177Lu) on the viability, cytotoxic function, and phenotype of third generation anti-GD2 CAR T cells." Or split in two sentences, for instance like this: "Herein, we evaluated in vitro the impact of radiation dose on the viability, cytotoxic function, and phenotype of third generation anti-GD2 CAR T cells. The radiation dose was delivered by actinium-225 (225Ac), an α-particle emitter, or by lutetium-177 (177Lu), a β-particle emitter."
Response: We thank the reviewer for this suggestion and have rephrased the closing sentence as suggested.
- Only the radiation doses (1, 2, or 6 Gy) are reported in the paper, not the activities (Bq). It will be of interest to the reader also to know what activities are used. This includes the use of 90Y briefly mentioned in the subsection "In vitro dosimetry".
Response: We thank the reviewer and have now included the specific activity for each radiation dose of radionuclide, see Methods and Materials sections.
- The irradiation is performed on a 6-well plate. While the wells are physically separated, they will not be completely separated regarding radiation. Cross-irradiation among wells is likely a minor or very minor effect, but please address it, for instance by explaining why the effect is so small it can be ignored, if this is the case.
Response: In the experimental set up as can be seen in Fig 2 and 3, on the 6-well plate, we staggered the wells used for the experiments to increase the distance between wells and minimize cross-irradiation among wells. Furthermore, the range of emitted particle is relatively small compared to the distance between the wells.
- Figures 1, 2, and 3 use very small print. It is suggested to enlarge the figures to make the font size comparable to that of Figure 4.
Response: Font size in Fig 1-3 have been increased.
- Figure 3 appears to be missing a column for 2Gy-irradiation using 225Ac (Fig. 2 has Ac before Lu, Fig. 3 has Lu before Ac). Furthermore, is it by intention that the order and notation differs slightly from Figure 2? (Fig 2 writes "225Ac (1 Gy)", Fig. 3 writes "225Ac-1Gy")
Response: We thank the reviewer for bringing this to our attention. The cytotoxic activity of CAR T cells after 2 Gy irradiation with 225Ac was not performed because there were not enough viable CAR T cells after the irradiation to perform a triplicate experiment. This was clarified in the legend of Fig.3. The order and notation have been corrected.
- By the end of the Results section, non-significant "trends" in Figure 4 are described. Please avoid describing "trends" when the results are not significant. In fact, the text on "trends" is counter to the legend of Figure 4, which opens with the words: "225Ac or 177Lu DOES NOT IMPACT the expression ..."
Response: We’ve corrected this terminology as suggested.
- In the Discussion, the difference between alpha- and beta-irradiation is discussed. It is well-known that alpha particles have a higher biological effectiveness than beta particles (and photons), but the present formulations (page 7) can be read as if this was a new finding: "This difference is biological effect is likely attributable to the linear energy transfer (LET) difference between these two radionuclides." Please make it more clear that the difference is not a new result, e.g. by including a reference to a work on relative biological effectiveness. (If in doubt where to find a good reference, take a look at ICRP Publication 92, which can be freely downloaded from https://icrp.org/page.asp?id=5 ).
Response: We have clarified this point in the discussion section and included an appropriate citation.
- On page 7, the meaning of this phrase is unclear: "... our in vitrodosimetry enabled the comparison of a similar dose of radiation delivered by a radionuclide instead of activity." It is not clear how "radionuclide" opposes "activity" ("radionuclide instead of activity"). Is the meaning that dosimetry enabled comparison with the actual radiation dose (Gy) rather than with the activity (Bq) used? Please rephrase.
Response: We have rephrased this sentence for added clarification.
- The authors suggest that their findings may be used to combined CAR T cell therapy with TRT, to irradiate the CAR T cells in vivoby the activity from the radioactivity given as TRT. Surely an idea worth further studies. Have the authors considered the possibility of in vitro irradiation of CAR T cells to stimulate the CAR T cells before application of the CAR T cells to the patient? This might broaden the application to patients not given TRT, although it still includes the handling of radioactivity by the staff. If a similar effect could be obtained with photon irradiation of the CAR T cells, it might be performed with easy-accessible radionuclides like 99mTc, or even without radioactivity, using an x-ray device. These are just ideas for thought.
Response: We thank the reviewer for these great ideas. While the increase in the cytotoxic activity of CAR T cells observed after irradiation can be beneficial for therapy, such irradiation is ultimately detrimental to their viability. It may be possible to identify very low doses of radiation delivered by EBRT preferably to stimulate CAR T cell killing function with minimal effects on viability.
VARIOUS / VERY MINOR
- In the first paragraph of the introduction, part of a sentence seems to be missing: "Structurally, a CAR is composed of ... and (Fig. 1A)." - and what? (co-stimulatory domains?)
Response: This has been corrected as suggested.
- Later in the Introduction: "Thus, we postulated that TRT may help ..." Is "postulated" = claimed that it was so, the right word here? Suggestion: "Thus we propose that TRT may help ..." (the change from past to present tense in this place is part of the suggestion - for the reader, past tense would naturally refer to earlier work.)
Response: This has been corrected as suggested.
- Bottom of page 2: There seems to be unwanted repetition in the phrase "... and how the timing the timing ..."
Response: This has been corrected as suggested.
- Middle of page 5: It is suggested to rephrase the sentence "The viability ... decreased to 6.96% ... compared to 52.2%." Suggestion: "The viability ... decreased from 52% to 7.0% ..." (the suggested fewer significant figures are based on the error bars in Figure 3.)
Response: This has been corrected as suggested.
- The abbreviation "PBS" is used but not defined.
Response: This has been corrected as suggested.
- Font/formatting: In a few places, the text font deviates from the standard. This seems to be in parts of the text where the Greek letters α and β are used.
Response: This has been corrected as suggested.